# Study on the Maximum Safe Instantaneous Input of the Steering Wheel against Rollover for Trucks on Horizontal Curves

**DOI:** 10.3390/ijerph19042025

**Published:** 2022-02-11

**Authors:** Jinliang Xu, Tian Xin, Chao Gao, Zhenhua Sun

**Affiliations:** 1School of Highway, Chang’an University, South 2nd Ring Road, Beilin District, Xi’an 710064, China; xujinliang@chd.edu.cn (J.X.); gaochao@chd.edu.cn (C.G.); 2Shaoxing Communications Investment Group Co., Ltd., No.135 Fenglin West Road, Jinghu District, Shaoxing 312000, China; hmfzj@163.com

**Keywords:** truck rollover, emergency steering, steering wheel instantaneous input, transient bicycle model, travel path, rollover margin, computer simulation

## Abstract

Truck rollover crashes on horizontal curves have been recognized as one of the most serious types of crashes. Driver’s instantaneous emergency steering maneuvers (DIESM) play an important role in truck rollover crashes, but have not received much attention. In the present study, the radius of curvature of the actual vehicle travel path (AVTP) under DIESM was calculated based on the transient bicycle model. Rollover margins were used to evaluate the truck-rollover potential under DIESM. To calculate rollover margins, the lateral acceleration under DIESM was calculated based on the radius of the curvature of the AVTP. A rollover threshold formula was introduced to calculate vehicle’s rollover thresholds by distinguishing two turning conditions. According to rollover margins, the maximum safe instantaneous input of the steering wheel against rollover for trucks was obtained. Moreover, theoretical results were verified by computer simulation. Results showed: (1) The maximum safe instantaneous inputs of the steering wheel were 259°, 212°, 182°, 162°and 147°, respectively, at speeds of 60 km/h, 70 km/h, 80 km, 90 km and 100 km when the superelevation rate was 0, and (2) superelevation significantly affected truck-rollover potential; the worst turning condition was turning from the inside to the outside of the curve. Due to the consideration of the wheelbase, the centroid position, the tire’s cornering stiffness and the suspension roll gain, the prediction results were more accurate.

## 1. Introduction

Truck rollover crashes are a major traffic safety concern due to high fatalities and serious injuries. In the United States, the proportion of all traffic fatalities associated with rollovers was about 15% in 2020 [1]. An analysis of 2020 crash data from the National Highway Traffic Safety Administration (NHTSA) revealed that trucks experienced the highest rollover rate in crashes [2]. This issue has received much attention by researchers and highway and vehicle agencies [3,4,5,6,7]. There have since been continuing efforts by researchers to reduce fatalities and serious injuries of rollovers [8,9,10,11].

Rollovers are generally associated with lateral acceleration and obstacle tripping [12]. The rollovers associated with lateral acceleration were analyzed in this study. When the lateral acceleration experienced by a vehicle exceeds that vehicle’s rollover threshold, it will cause the vehicles to overturn.

Horizontal curves have long been recognized to be among those road locations most vulnerable to skid and rollover occurrence [13]. However, horizontal curves designed in accordance with present design policy, even minimum-radius curves, have been shown to provide a reasonable margin of safety against skidding and overturning under normal circumstances for vehicles [14,15,16,17]; in particular, this research conclusion is based on the following idealized conditions: (1) vehicles are simplified as a point-mass model, (2) vehicles travel at the speed not exceeding the road design speed, (3) vehicles follow the horizontal curve exactly, i.e., the radius of curvature of the actual vehicle travel path (AVTP) is equal to that of the horizontal curve. However, temporary disturbances to the driver–vehicle system were not involved in the above research, such as throwing objects from the front vehicle, a sudden traffic accident, obstructed vision, etc., and driver’s emergency maneuvers, especially, driver’s instantaneous emergency steering maneuvers (DIESM) and emergency braking maneuvers are needed. DIESM will make vehicles no longer maintain the roadway design path. It easily leads to operation errors and poses a higher rollover potential [18,19]. Some studies have shown that the complex vehicle dynamics models (i.e., the transient bicycle model) indicate that the point-mass model generally overestimates the margins of safety against skidding and overturning when driver behaviors (i.e., steering and braking) change the forces on trucks [20,21,22].

To study the rollover stability of trucks and to investigate the force and dynamic response of the vehicle during driving, vehicle experts had established many vehicle dynamics models in previous studies [23,24]. Some models were established based on classical Newton mechanics theory, others were based on multibody-system-dynamic theory. The complexity of the model varies with the theory of modeling. The selection of the model will require tradeoffs between simplicity, accuracy, and ease of use. A more complex model means that a larger number of parameters are needed with higher order differential equations. For the consideration of comprehensive factors, we selected a vehicle dynamics model with three degrees of freedom (DOF) in the present study.

The main purpose of this study was to quantify the effect of DIESM on the truck rollover potential and obtain the maximum safe instantaneous input of the steering wheel against rollover for trucks on horizontal curves. We performed quantitative analyses based on theoretical considerations. Rollover margin was used to evaluate truck rollover potential under DIESM. Firstly, the transient bicycle model was used to develop the mathematical relationship between the radius of curvature of the AVTP and the steering wheel instantaneous input, wheelbase, centroid position, speed and tire’s cornering characteristics were considered. The lateral acceleration experienced by vehicles under DIESM was calculated based on the mathematical relationship. Then, a vehicle’s rollover threshold formula considering suspension roll gain was introduced to calculate vehicle’s rollover thresholds by distinguishing two turning conditions. Finally, the rollover margin was the rollover threshold minus the lateral acceleration under DIESM; according to the rollover margin, the maximum safe instantaneous input of the steering wheel against rollover for trucks was obtained. Moreover, the accuracy of theoretical results was verified by computer simulation. This study provides useful insights for transportation engineers and policy makers who seek to identify relationships between emergency steering and truck rollover potential in order to improve driver training and education programs designed to modify risky driving habits. Furthermore, it provides a theoretical basis for the planning of obstacle-avoidance path of intelligent driving.

## 2. Literature Review

### 2.1. Effects of Horizontal-Curve Features on Rollover

The vehicle stability on the horizontal curve is affected by the lateral acceleration [25,26,27]. Design criteria for the horizontal curve are based on the lateral force coefficient μ (related to lateral acceleration); μ is a function of speed, curve radius and superelevation [17]. The study showed that the lateral acceleration was too large to make the driver feel comfortable while the vehicle traverses a sharp curve, so it increased the difficulty of vehicle control [25]. Some studies pointed out that when lateral acceleration was used to evaluate the potential for rollovers, horizontal-curve radius, superelevation and their interaction had significant effects on rollovers [26,27]. These studies show that rollovers are related to excessive lateral acceleration, and that superelevation and curve radius significantly affect the lateral acceleration.

Harwood and Mason assessed the safety of the road design policy, and they believed that the horizontal curve designed according to the current design policy could ensure that passenger cars and trucks do not overturn or skid as long as their speed did not exceed the road design speed in most situations [15]. However, current design policy for the horizontal curve is based on a mathematical model that represents the vehicle as a point mass. It has been shown that the complex vehicle dynamics models (i.e., the transient bicycle model and multibody models) indicate that the point-mass model generally overestimated the rollover margins of vehicles, especially for trucks [6,22]. These studies show that the point model may be insufficient in evaluating vehicle stability in special situations and that the vehicle stability should be analyzed with a complex model.

### 2.2. Effects of Vehicle Characteristics on Rollover

The studies showed that vehicle characteristics, especially, vehicle configuration and the type of loading cargo, directly affect the roll stability [28,29,30,31]. Gillespie and Verma suggested that special attention should be given to liquid-filled vehicles because the interaction between the liquid and the vehicle is more complicated; they found that the rollover threshold of liquid tankers was 0.36 g because of their higher CG (center of gravity) and different suspension systems [28]. Harwood et al. reported that the conservative value (worst-case) rollover threshold for trucks was approximately 0.35 g, and that the rollover threshold was about 85% of 0.35 g, i.e., 0.3 g when accounting for superelevation and suspension [16]. An investigation in the United States found that the frequency of rollover crashes of full-load trucks with a rollover threshold of 0.4 g was about 10 times higher than that of none-load trucks with a rollover threshold of 0.65 g [31].

These studies showed that CG height and suspension system characteristics importantly impacted the vehicle’s rollover threshold. Different suspension systems possess different roll characteristics. The roll stiffness of a truck suspension system is generally higher than that of a car suspension system [28]. Therefore, we cannot ignore the influence of CG height and suspension system on a vehicle’s rollover threshold.

### 2.3. Effects of Driver Behaviors on Rollover

Driver behaviors have been shown to influence vehicle rollovers [7,32,33]. Research efforts and crash data showed that more than 90% of crashes were related to unreasonable or wrong driving behavior; the more common behaviors are speeding, emergency steering and emergency braking [2,34,35].

Khattak et al. analyzed the effects of driver behaviors on truck rollovers and injury severity using three years of accident data from North Carolina; the results showed that emergency steering and steering movements were associated with higher potential for rollovers and higher injury severity [36]. Some studies showed that there were two collision avoidance behaviors of braking and steering in emergency conditions. Compared with braking collision avoidance, emergency steering collision avoidance required smaller longitudinal distance and showed higher collision avoidance efficiency under certain conditions. These conditions included high relative speed and low adhesion coefficient [37]. For high-speed vehicles, TTC (time to collision) < 2 s is the screening condition for whether the driver uses emergency steering (emergency lane change) [38]. However, emergency steering was shown to be a driving behavior that could easily lead the driver to make mistakes and cause rollover crashes [18,19].

By summarizing the previous studies, there are several main approaches to investigate the effects of driver behavior on rollover, which are the approaches of accident statistics analysis, driving simulation and theoretical modeling [32,33,34,35,36,37,38]. These approaches contain their own characteristics and limitations. For the approach of accident statistics analysis, it cannot explain the relationship between rollover crashes and various driver behaviors in detail, which makes it difficult to evaluate the relationship between driver behavior and rollover crashes. For the approach of driving simulation, the knowledge that the driving is under test conditions causes the driver to wait for something to happen and subjects appear to feel that they can’t be harmed, which is very different from normal driving conditions. For the approach of theoretical modeling, it can quantify the relationship between driver behavior and vehicle rollovers, however, how to choose an appropriate vehicle dynamic model is the key issue.

Previous studies showed that rollover crashes are associated with emergency steering behavior and that the approach of theoretical modeling holds great advantages in quantitative analyses. Therefore, this study used a three-DOF model to measure the effects of DIESM on truck rollovers.

## 3. Theoretical Analysis Methods

Rollover margin was an important index to evaluate rollover potential in previous studies [22,39]. Rollover Margin was defined in two ways: (1) one rollover margin was based on lateral acceleration, which represented the difference between the present lateral acceleration and the maximum lateral acceleration that a vehicle could experience without overturning; (2) rollover margin was also defined by the proximity of the load-transfer ratio to an absolute value of unity, e.g., how close an axle was to experiencing wheel lift. In both cases, a value of zero indicated the onset of wheel lift [22].

Based on the second definition, each tire of the vehicles will be regarded as a separate kinematic unit, i.e., the vehicle will be regarded as a multibody model, and all the forces of each tire will be considered [39]. Considering the complexity of the influence factors and difficulty in solving the multibody model, the present study adopted the rollover margin based on lateral acceleration as the evaluation index of rollover potential. As a result, it is necessary to ascertain the lateral acceleration under DIESM and the maximum lateral acceleration that a vehicle can experience without overturning, which is called the rollover threshold. To show the evaluation process clearly, a flow-process diagram is shown in Figure 1.

### 3.1. Transient Bicycle Model

An SAE axis system is generally used to describe vehicle motion, as shown in Figure 2. This axis system (xyz) is fixed to the vehicle (body) with z-axis pointing downwards. The origin is located at vehicle CG (center of gravity). Inertia values relative to xyz coordinate frame are assumed to be constant. The motion of the vehicle on the xoy plane is analyzed in this section. To facilitate the analysis, it is necessary to simplify the vehicle, and a brief introduction of the principle of vehicle steering is showed.

#### 3.1.1. Basic Assumptions

A common simplification in vehicle dynamics is assumed for this analysis: the vehicle is idealized as a rigid beam, and each axle is represented as a single tire situated at the midline of the vehicle. The resulting model is termed a “bicycle model”, a classic vehicle dynamics model. Assumptions made for the bicycle model are as follows:The vehicle is right/left symmetric;There is no acceleration in the longitudinal direction of the vehicle, and the longitudinal speed is constant;The steering input is applied to the front wheel through the steering wheel with a fixed transmission ratio;Aerodynamics and rolling resistance of the tires are ignored.

Through doing this, the motion of the vehicle on the xoy plane is described by two-DOF, namely lateral translation and yaw motion, as shown in Figure 3.

#### 3.1.2. Principle of Vehicle Steering

There are three typical states during vehicle steering, shown in Figure 4.

State1: vehicles travel in a straight line.

State2: Vehicles make a stable circular motion around the curvature center O, in which the turning radius of the vehicle is equal to circular-curve radius R. 

State3: drivers take instantaneous emergency steering in unexpected situation. In turn, front-tire steering angle increases from $\delta_{V}$ to $\delta_{V}\prime$ sharply, the curvature center becomes O′, and the turning radius decreases from R to ρ greatly, which will induce a large increase in lateral acceleration, especially for trucks at high speeds.

### 3.2. Radius of Curvature of the AVTP under DIESM

As the vehicle travels at high speeds (speeds above 60 km/h) and its lateral acceleration is more than 0.4 g, the tire slip greatly impacts travel paths, so the nonlinear characteristics of the vehicle need to be considered [40]. According to the geometric relationship in Figure 5, one obtains:(1)$$\rho =l\frac{\sin\left[90{}^{\circ}-\left(\delta_{V}-\alpha_{V}\right)\right]}{\sin\left[\delta_{V}-\left(\alpha_{V}-\alpha_{H}\right)\right]}\;\;\;\;\;$$
where: *ρ*—radius of curvature, m; l—wheelbase, $\delta_{V}$—front-tire steering angle, °;$\;\alpha_{V}$—front-tire slip angle, °;$\;\alpha_{H}$—rear-tire slip angle, °.

Because $\delta_{V}-\alpha_{V}\;$ is much less than 90°, then $\sin\left[90{}^{\circ}-\left(\delta_{V}-\alpha_{V}\right)\right]\approx \sin90{}^{\circ}=1$, and the front-tire steering angle and the tire’s slip angle are very small, use small angle approximation,$\;\sin\left[\delta_{V}-(\alpha_{V}-\alpha_{H})\right]\approx \pi /180\cdot \left[\delta_{V}-\alpha_{V}-\alpha_{H}\right]$. Then Equation (1) can be expressed as:(2)$$\rho \approx \frac{l}{\delta_{V}-\left(\alpha_{V}-\alpha_{H}\right)}\cdot \frac{180}{\pi }$$

The tire’s slip angle can be expressed as:(3)$$\alpha_{V}=\frac{F_{yV}}{c_{\alpha_{V}}}=\frac{1}{c_{\alpha_{V}}}\frac{b}{l}m\frac{v^{2}}{\rho }\;\cdot \frac{180}{\pi }$$
(4)$$\alpha_{H}=\frac{F_{yH}}{c_{\alpha_{H}}}=\frac{1}{c_{\alpha_{H}}}\frac{a}{l}m\frac{v^{2}}{\rho }\;\cdot \frac{180}{\pi }$$
where:$\;F_{yV}$—lateral force of front axle, N;$\;F_{yH}$—lateral force of rear axle, N;$\;c_{\alpha_{V}}$—front-tire cornering stiffness, rad/N;$\;c_{\alpha_{H}}$—rear-tire cornering stiffness, rad/N; v—speed, m/s; m—vehicle mass, kg; a—distance from centroid to front axle, m; b—distance from centroid to rear axle, m.

Therefore,
(5)$$\alpha_{V}-\alpha_{H}=\frac{c_{\alpha_{H}}b-c_{\alpha_{V}}a}{c_{\alpha_{V}}c_{\alpha_{H}}l}m\frac{v^{2}}{\rho }\;\cdot \frac{180}{\pi }$$

And:(6)$$\delta_{V}=\frac{\delta_{L}}{i_{L}}$$
where: $\delta_{L}$—steering wheel instantaneous input, °; $i_{L}$—steering ratio.

Substituting Equations (5) and (6) into Equation (2), one obtains:(7)$$\rho =\frac{l}{\frac{\delta_{L}}{i_{L}}-\left(\frac{c_{\alpha_{H}}b-c_{\alpha_{V}}a}{c_{\alpha_{V}}c_{\alpha_{H}}l}m\frac{v^{2}}{\rho }\;\cdot \frac{180}{\pi }\;\;\right)}\cdot \frac{180}{\pi }$$

Then, ρ can be expressed as:(8)$$\rho =\frac{i_{L}\left(l+\frac{c_{\alpha_{H}}b-c_{\alpha_{V}}a}{c_{\alpha_{V}}c_{\alpha_{H}}l}mv^{2}\right)\;}{\delta_{L}}\cdot \frac{180}{\pi }$$

Equation (8) shows that the radius of curvature of the AVTP (ρ) is not only related to the wheelbase (l), the steering wheel instantaneous input ($\delta_{L}$) and steering ratio ($i_{L}$), but also to the speed (v), centroid position (a,b) and tire’s cornering stiffness ($c_{\alpha_{V}},c_{\alpha_{H}}$) when vehicles travel at high speeds. If vehicles and loading condition are determined, l, $i_{L}$, a, b, $c_{\alpha_{V}}$, $c_{\alpha_{V}}$ are determined. Then, the radius of curvature of the AVTP (ρ) depends on the steering wheel instantaneous input ($\delta_{L}$) and speed (v). It is easy to find that there is a negative correlation between the radius of curvature of the AVTP (ρ) and steering wheel instantaneous input ($\delta_{L}$), the relationship between the radius of curvature of the AVTP (ρ) and the speed (v) depends on the positive or negative of “$\left(c_{\alpha_{H}}b-c_{\alpha_{V}}a\right)$” and it requires further quantitative calculation.

### 3.3. Rollover Margins

#### 3.3.1. Vehicle’s Lateral Acceleration under DIESM

According to Newton’s law, the lateral acceleration under DIESM can be expressed as:(9)$$a_{y}=\frac{v^{2}}{\rho }$$
where: $a_{y}$—centrifugal acceleration, m/s^2^.

Substituting Equation (8) into Equation (9), the lateral acceleration under DIESM can be expressed as:(10)$$a_{y}=\frac{\pi \delta_{L}}{180i_{L}\left(\frac{l}{v^{2}}+\frac{c_{\alpha_{H}}b-c_{\alpha_{V}}a}{c_{\alpha_{V}}c_{\alpha_{H}}l}m\right)\;}$$

Equation (10) shows that ① the larger the steering wheel instantaneous input ($\delta_{L}$), the larger is the lateral acceleration ($a_{y}$), ② the higher the driving speed (v), the larger is the lateral acceleration ($a_{y}$). It conforms to the basic rule and also quantifies the steering wheel instantaneous input ($\delta_{L}$) and the lateral acceleration ($a_{y}$).

#### 3.3.2. Vehicle’s Rollover Threshold

In this study, the vehicle system contained three degrees of freedom in total, namely: lateral translation and yaw and roll motion. In addition to the mentioned two degrees of freedom on the xoy plane, there was also a roll degree of freedom of movement on the yoz plane, as shown in Figure 6. The motion of the vehicle on the yoz  plane is analyzed in this section. Assumptions were made as follows:The roll angle and roll angular velocity of the unsprung mass are ignored;The movement of the vehicle in the *z*-axis caused by uneven road surface is ignored.

On the yoz  plane, the vehicle contains a movement limit, i.e., rollover limit, which is related to the vehicle’s rollover threshold. Vehicle’s rollover threshold refers to the maximum lateral acceleration that a vehicle can experience without overturning, which is mainly related to the CG height, track width and suspension characteristics of vehicles [41,42].

Torbic et al. gave a vehicle’s rollover threshold formula considering suspension and superelevation effects [22], as shown in Figure 7:(11)$$\mu_{y,rollover}=\frac{a_{y}}{g}=\frac{\frac{T}{2h_{g}}+i_{h}}{1+\left(1-\frac{h_{r}}{h_{g}}\right)R_{\varphi }}\;$$
where:$\;\mu_{y,rollover}$—rollover threshold, g; g—gravitational acceleration, m/s^2^; T—track width, m;$\;h_{g}$—CG height, m;$\;i_{h}$—superelevation rate;$\;h_{r}$—distance from roll center to ground, m; φ—roll angle,$\;\varphi =R_{\varphi }\cdot (a_{y}/g)$,°;$\;R_{\varphi }$—a roll gain, rad/g;$\;a_{y}/g$—normalized acceleration within curve.

To develop approximate estimates of vehicle’s rollover threshold, an approximate value of $R_{\varphi }$ = 0.17 rad/*g* was assumed given the fact that most vehicles exhibit approximately 1° of roll per 0.1 *g* of lateral acceleration (10°/*g* corresponds to 0.17 rad/*g*) [41]. A worst-case value of $h_{r}/h_{g}$ would be zero. Therefore, Equation (11) can be rewritten as:(12)$$\mu_{y,rollover}=\frac{a_{y}}{g}\approx \frac{1}{1.17}\left(\frac{T}{2h_{g}}+i_{h}\right)\approx 0.85\left(\frac{T}{2h_{g}}+i_{h}\right)$$

Drivers will inevitably encounter all kinds of emergencies, so they must take an emergency operation such as emergency steering to avoid danger; vehicles may turn from the outside to the inside of the curve, or from the inside to the outside of the curve, as shown in Figure 8. The turning condition determines the positive and negative of superelevation, which, in turn, affects the vehicle’s rollover threshold. Therefore, the vehicle’s rollover threshold was calculated by distinguishing two turning conditions:If turning from the outside to the inside of the curve, the vehicle’s rollover threshold can be written as Equation (12).If turning from the inside to the outside of the curve, the vehicle’s rollover threshold can be written as:
(13)$$\mu_{y,rollover}=\frac{a_{y}}{g}=0.85\left(\frac{T}{2h_{g}}-i_{h}\right)$$

Equation (13) shows that the superelevation does not improve the vehicle’s roll stability but increases the potential for rollovers under the second condition; the greater the road superelevation rate ($i_{h}$), the smaller is the vehicle’s rollover threshold ($\mu_{y,rollover}$).

According to the definition of the rollover margin, it was also calculated by distinguishing two turning conditions:Turning from the outside to the inside of the curve.
(14)$$RM=0.85\left(\frac{T}{2h_{g}}+i_{h}\right)-\frac{a_{y}}{g}$$
where: RM—rollover margin, g.

Substituting Equation (10) into Equation (14), one obtains:(15)$$RM=0.85\left(\frac{T}{2h_{g}}+i_{h}\right)-\frac{\pi \delta_{L}}{180i_{L}\left(\frac{l}{v^{2}}+\frac{c_{\alpha_{H}}b-c_{\alpha_{V}}a}{c_{\alpha_{V}}c_{\alpha_{H}}l}m\right)\;g\;\;}$$

Equation (15) shows ① the larger the steering wheel instantaneous input ($\delta_{L}$), the closer is the present lateral acceleration to the vehicle’s rollover threshold, i.e., the smaller is the rollover margin (RM) and the higher is the potential for rollovers, ② the higher the driving speed, the smaller is the rollover margin and the higher is the potential for rollovers.

2.Turning from the inside to the outside of the curve.


(16)
$$RM=0.85\left(\frac{T}{2h_{g}}-i_{h}\right)-\frac{a_{y}}{g}$$


Substituting Equation (10) into Equation (16), one obtains:(17)$$RM=0.85\left(\frac{T}{2h_{g}}-i_{h}\right)-\frac{\pi \delta_{L}}{180i_{L}\left(\frac{l}{v^{2}}+\frac{c_{\alpha_{H}}b-c_{\alpha_{V}}a}{c_{\alpha_{V}}c_{\alpha_{H}}l}m\right)\;g\;\;}$$

Comparing Equation (15) with Equation (17), in addition to the same findings, it can be found that the rollover threshold is smaller under the condition of turning from the inside to the outside of the curve, the rollover margin (RM) is easier to reach the zero point, and the vehicle is easier to overturn, so this turning condition is the worst for driving.

### 3.4. The Maximum Safe Instantaneous Input of the Steering Wheel against Rollover

#### 3.4.1. Formula Deduction

The limit condition for vehicles to travel without overturning is RM=0. According to this limit condition, we can solve the maximum safe instantaneous input of the steering wheel against rollover under DIESM (shortened to the maximum safe input of the steering wheel) by distinguishing two turning conditions:Turning from the outside to the inside of the curve.
(18)$$\delta_{Lmax}=\frac{19431li_{L}}{\pi V^{2}}\left(\frac{T}{2h_{g}}+i_{h}\right)+477.139m\;i_{L}\;\left(\frac{T}{2h_{g}}+i_{h}\right)\frac{c_{\alpha_{H}}b-c_{\alpha_{V}}a}{c_{\alpha_{V}}c_{\alpha_{H}}l}$$
where: V—speed, km/h;$\;\delta_{Lmax}$—the maximum safe input of the steering wheel, °.

2.Turning from the inside to the outside of the curve.


(19)
$$\delta_{Lmax}=\frac{19431li_{L}}{\pi V^{2}}\left(\frac{T}{2h_{g}}-i_{h}\right)+477.139m\;i_{L}\;\left(\frac{T}{2h_{g}}-i_{h}\right)\frac{c_{\alpha_{H}}b-c_{\alpha_{V}}a}{c_{\alpha_{V}}c_{\alpha_{H}}l}$$


Equations (18) and (19) show if the vehicle is determined, steering ratio ($i_{L}$), wheelbase (l), track width (T) are determined. With the change of vehicle loading conditions, the vehicle mass (m), centroid position (a, b) and CG height ($h_{g}$) and tire’s cornering stiffness ($c_{\alpha_{V}},c_{\alpha_{H}}$) also vary; the maximum safe input of the steering wheel ($\delta_{Lmax}$) is closely related to the driving speed (V) and the vehicle’s rollover threshold. Further analysis shows that the maximum safe input of the steering wheel ($\delta_{Lmax}$) is negatively correlated with the driving speed (V), CG height ($h_{g}$), and the relationship with superelevation depends on turning conditions. If the vehicle turns from the outside to the inside of the curve, it is positively correlated with superelevation rate ($i_{h}$), but if the vehicle turns from the inside to the outside of the curve, it is negatively correlated with superelevation rate ($i_{h}$). These results are quite satisfactory because they are consistent with the previous qualitative research results [26,27,43], but they can also quantitatively describe the relationship between DIESM and vehicle rollovers.

#### 3.4.2. Identification of Parameters

To calculate the maximum safe input of the steering wheel, tire’s cornering stiffness ($c_{\alpha_{H}}$, $c_{\alpha_{v}}$) needs to be solved. Figure 9 shows the tire curves at five different loads for a truck tire. These cornering stiffness values were obtained from tire curves, which are taken from data sets for truck tires of TruckSim.

In the case of a small slip angle, the tire’s lateral force is approximately proportional to the tire’s slip angle [44]. The slope of the tire curves is the cornering stiffness of a tire in the area of a small slip angle. The “magic formula” Equation (20) was used to calculate the slope of the curve at zero slip angle:(20)$$y=D\sin(C\tan^{-1}(Bx-E(Bx-\tan^{-1}Bx)))$$

The product of coefficients *B*, *C* and *D* in Equation (20) corresponds to the slope of the curve at zero slip angle, i.e., *BCD* = tan θ [45]. According to Equation (20), the tire curves were fitted with Matlab and three coefficients *B*, *C* and *D* were obtained.

The detailed process is as follows: the first row of the table represents different vertical loads and the first column represents tire’s slip angle in Figure 9. Firstly, compare tire’s actual vertical load with the data in the first row of the table; then, select the closest tire curve and import the data of that curve into Matlab, and then fit it according to Equation (20); finally, three coefficients *B*, *C* and *D* were obtained.

## 4. Computer Simulation Methods

Computer simulation has been an efficient way to study vehicle stability because of the difficulty and danger in field tests of skid and rollover crashes [46]. With more than twenty years of real-world validation, TruckSim is the universally preferred tool for analyzing vehicle dynamics and calculating a truck’s performance characteristics. It is used worldwide by over 110 OEMs and Tier 1 suppliers and over 200 universities and government research labs [47].

In this study, Trucksim2016.1 was used to simulate the dynamics and kinematics of trucks.

The simulation purpose was to test the maximum safe input of the steering wheel under full-load condition.

### 4.1. Simulation Modeling

Simulation modeling mainly includes vehicle modeling, road modeling and driver behavior modeling. In vehicle modeling, the vehicle prototype was a FAW Jiefang J6M8X4 series truck, which was selected because it accounted for a large proportion of current trucks in China, because of their operational characteristics, especially the fact that they are prone to overturn [48]. In road modeling, the number of lanes was set as a two-lane, the width of each lane was set to 3.75 m with hard shoulders on both sides, and roadside was set as a lawn [49]. The specific parameters of vehicle modeling and road modeling can be viewed in the following simulation scenes. This section focuses on the method of driver behavior modeling, which is involved in setting the target speed and the establishing travel paths. They are described in detail below.

#### 4.1.1. Target Speeds and Travel Paths

To simulate DIESM, this study set a variety of target speeds and travel paths with TruckSim. Vehicles followed the given path at a target speed. As the path is given, steering information is given. Drivers receive steering information and turn the steering wheel according to the steering information.

Target speeds in this study were set as 100 km/h, 90 km/h, 80 km/h, 70 km/h and 60 km/h.

Travel paths drive the vehicle to change lane from the current lane to the adjacent lane, as shown in Figure 10. The details are as follows: the vehicle launches at zero time and travels along the centerline of the current lane; now, the ordinate (lateral offset) is −1.875. After traveling for a short time, all vehicle’s kinematics indexes reach stable levels; the vehicle begins to take a lateral shift gradually. Finally, the vehicle reaches to the centerline of the adjacent lane; now, the ordinate (lateral offset) is 1.875. The total lateral offset distance is one-lane width (3.75 m); lane change maneuvers are ended. The abscissa represents the road station information, from which it is easy to calculate the longitudinal driving distance. Travel paths were numbered from 1 to 9.

#### 4.1.2. Methods of Establishing Travel Paths

The mathematical analysis in the previous section showed that steering (steering wheel instantaneous input) affected the travel path, which in turn affected the lateral acceleration and vehicle stability. Based on the corresponding relationship between steering and the travel path, we simulated various steering by establishing various travel paths, and observed the transient response of vehicles.

To show the methods clearly, a flow-process diagram is given shown in Figure 11.

Path 1 is a general lane-change path (a safe pattern of lane changes frequently taken by most drivers under normal conditions). Path 1 referred to the example data sets of TruckSim. To simulate DIESM, emergency lane-change paths were built. Compared with the general lane change, the emergency lane change is to complete the same lateral offset in a shorter time. We tried to build path 2 (an emergency lane-change path) based on path 1. Specifically, we set the ordinate (lateral offset) of path 2 equal to that of path 1, and set the abscissa (station) of path 2 equal to that of path 1 multiplied by a coefficient k (k<1), which was equivalent to shortening lane-change time.

Time of an emergency lane change.

According to current standards, for high-speed vehicles, TTC (time to collision) < 2 s is the screening condition for whether the driver takes an emergency lane change [38]. The relationship between steering, vehicle location on the roadway and time during lane-change maneuvers is briefly shown in Figure 12a. Time1 to time 4 represents the total time taken to complete a lane change, marked as t1, time1 to time2 represents the time to go from steering initiation to the maximum, marked as t2. A simple relationship in Figure 13 shows t1=TTC (vehicle length is ignored), so t1 was controlled below 2 s in emergency lane-change paths.

2.Determination of the coefficient k

① Given V = 100 km/h, t1=TTC=2 s, S (longitudinal driving distance) shown in Figure 13 was calculated; the longitudinal driving distance of path 1 was marked as  S′; the coefficient k=S/S′≈0.5. Substituting the coefficient k, a new path was built, and it was used to test the worst condition (full-load truck, V = 100 km/h) in this study. If the truck did not overturn following the new path under the worst conditions, it meant that the truck would not overturn following the new path under any other condition in this study. The test result showed that the truck could change lanes safely. Then, we tried to make k=0.4; another new path was built and was tested (the test method was the same as above), the test result showed that the truck could not complete lane change. Finally, the path (k=0.5) was select as path 2. When a truck travels at 100 km/h along the path 2, t2 is below 1 s. In this way, simulations realistically replicate DIESM.

② Given V = 60 km/h, t1=TTC=2 s, in the same way, the coefficient  k was calculated, k=S/S′≈0.25. Substituting the coefficient k, a new path was built, and the new path was used to test the condition (full-load truck, V = 60 km/h). The test result showed that the truck could change lanes safely. Then, we tried to make k=0.15; another new path was built and was tested (the test method was the same as above); the test result showed that the truck rolled over. Finally, the path (k=0.15) was selected as path 9.

By interpolating different k between k=0.15 and k=0.5 to make the difference between the adjacent paths as small as possible, paths 3~8 were built. The difference of time to completing the adjacent paths at 100 km/h and 60 km/h is 0.216 s and 0.360 s respectively. Such a small difference ensures the accuracy of the data.

Mathematical analysis in the previous section also showed that when the steering wheel input reaches the maximum; the lateral acceleration experienced by the truck also theoretically reaches the maximum at the same time, as shown in Figure 12a,b. To avoid the interference of potential factors on the results, it is specially noted that the maximum steering wheel input, the maximum lateral acceleration experienced by the truck and the transient response of the truck all refer to the value at time2 in the following tests.

### 4.2. Simulation Scenes Constructing

To test the maximum safe input of the steering wheel under full-load conditions, a number of simulation tests were carried out. The simulation plan was as follows: a full-load truck travelled at 100 km/h and completed lane change along the path numbered from No.2 to No.9 in turn; once the rollover occurred, the tests were over. Then, the maximum input of the steering wheel in the penultimate test was the maximum safe input of the steering wheel. Then, we changed the speed to 90 km/h, 80 km/h, 70 km/h and 60 km/h; in the same way, the maximum safe input of the steering wheel at different speeds could be obtained.

Based on the simulation purpose and simulation plan, 40 scenes were constructed. Simulation scenes constructing mainly involved vehicle elements, road elements and driver behavior elements. The description of these elements in the 40 scenes are as follows.

Vehicle elements: the total mass of the vehicle is 30,000 kg, the CG height of load is 2 m.

Road elements: curve radius is 5500 m, superelevation rate is 0, longitudinal grade is 0, tire-pavement friction coefficient is 0.7.

Driver behavior elements are shown in Table 1.

## 5. Results

### 5.1. Theoretical Results

To illustrate the validity of the theoretical formulae, this study takes the FAW Jiefang J6M8 × 4 series truck as an example, and performs calculations under different combinations of variables.


m=30,000 kg,  a=3.60 m,  b=4.25 m,  T=1.847 m, $\;h_{g}=1.79\;m$, $\;c_{\alpha_{v}}=361,749\;N/\mathrm{rad}$, $\;c_{\alpha_{H}}=441,600\;N/\mathrm{rad}$. The relationship between speed, superelevation rate and the maximum safe input of the steering wheel is shown in Figure 14.


Figure 14 shows that the maximum safe input the of steering wheel decreases significantly with the increase of speed. e.g., under the condition that the superelevation rate is 0.10 and the speed increases from 60 km/h to 100 km/h; if the truck turns from the outside to the inside of the curve, the maximum safe input of the steering wheel decreases from 309° to 176°; however, under the same condition, if the truck turns from the inside to the outside of the curve, the maximum safe input of the steering wheel decreases from 208° to 119°.

Figure 14 also shows that if the truck turns from the outside to the inside of the curve at a constant speed, the maximum safe input of the steering wheel increases significantly with the increase of superelevation rate. e.g., as the speed is 100 km/h and the superelevation rate increases from 0 to 0.10, the maximum safe input of the steering wheel increases from 147° to 176°. On the contrary, if the truck turns from the inside to the outside of the curve at a constant speed, the maximum safe input of the steering wheel decreases significantly with the increase of superelevation rate. e.g., under the same condition, and maximum safe input of the steering wheel decreases from 147° to 119°.

2.m=30,000 kg,  a=3.60 m,  b=4.25 m,  T=1.847 m, $\;c_{\alpha_{v}}=361,749\;N/\mathrm{rad}$, $\;c_{\alpha_{H}}=441,600\;N/\mathrm{rad}$, $\;i_{h}=0.06$. The relationship between speed, CG height and maximum safe input of the steering wheel is shown in Figure 15.

Figure 15 shows that the maximum safe input of the steering wheel decreases significantly with the increase of CG height. e.g., under the condition that the speed is 100 km/h and the CG height increases from 2 m to 3 m, if the truck turns from the outside to the inside of the curve, the maximum safe input of the steering wheel decreases from 149° to 105°. Meanwhile, under the same condition, if the truck turns from the inside to the outside of the curve, the maximum safe input of the steering wheel decreases from 115° to 71°.

### 5.2. Simulation Results

As the full-load truck travelled along the No. 8 path at 60 km/h, and the steering wheel instantaneous input reached 272°, a very short time later, the vertical forces of the left tires (L1, L2, L3 and L4) of the truck were zero, that is, all the left tires lifted and the truck was just about to overturn, as shown in Figure 16 and Figure 17. As the full-load truck travelled along the No. 9 path at 60 km/h, the truck overturned. Therefore, the maximum safe input of the steering wheel for a full-load truck at 60 km/h is 272°.Similarly, the other data were obtained. i.e., the maximum safe inputs of the steering wheel were 225°, 180°, 155° and 143° respectively for a full-load truck at 70 km/h, 80 km/h, 90 km/h and 100 km/h, as shown in Figure 16 and Figure 17b–e.

The three events, the steering wheel input reaching the maximum, the lateral acceleration reaching the maximum and the vertical forces of one side tires being all zero, did not appear at the same time, but the latter two lagged behind, which is due to the buffering effects of the suspension system.

### 5.3. Comparison of Theoretical and Simulation Results

It was found that the difference of maximum safe inputs of the steering wheel based on theory and simulation is not obvious; the biggest difference is 6.1%, as shown in the test Table 2. Obviously, the theoretical calculation of the maximum safe input of the steering wheel in this study showed a good degree of accuracy.

Analysis of results: the difference between the steering wheel instantaneous input ($\delta_{L}$) and the front-tire steering angle ($\delta_{V}$) comes from two aspects, i.e., steering ratio ($\;i_{L}$) and the stiffness of the steering system ($C_{L}$). For $\delta_{L}{}^{*}$($\delta_{L}{}^{*}=\delta_{L}/i_{L}$); it is necessary to consider the tire’s cornering stiffness and the stiffness of the steering system; for $\;\delta_{V}$, only the tire’s cornering stiffness needs to be considered [40]; however, for the convenience of calculation, we treat $\delta_{L}{}^{*}$ and $\;\delta_{V}$ as equal, ignoring the deformation of the steering system. Therefore, it leads to the deviation between theoretical and simulation results.

## 6. Discussion

Previous studies showed that horizontal curves designed according to the current design policy based on the point-mass model provides reasonable safety margins against rollovers and skids for vehicles in most situations [16,17]. Other studies have found that the point-mass model generally overestimated the margin of safety against rollovers and skids, and that the margin of safety cannot be guaranteed in unexpected situations when emergency steering or emergency braking is needed [20,21,22,50]. The present study confirms these findings, and additionally, supports the conclusions of Torbic et al. that steering maneuvers affect rollover margins [22]. This research finding showed that DIESM brings about the change of radius of curvature of the AVTP, which in turn significantly affect lateral acceleration and reduce rollover margins obviously. However, this change has usually been ignored. Specifically, our results showed that DIESM and driving speed significantly affect rollover margins; the larger the steering wheel instantaneous input, the higher is the potential for rollovers; the higher the driving speed, the higher is the potential for rollovers. For example, the speed is 100 km/h, when the instantaneous input of the steering wheel increases from 100° to 176°, the rollover margin decreases from 0.22 g to 0; while the instantaneous input of the steering wheel is 176°, when the speed increases from 60 km/h to 100 km/h, the rollover margin decreases from 0.22 g to 0 (the condition is that the superelevation rate is 0.10 and the truck turns from the outside to the inside of the curve). Moreover, computer simulation verified the accuracy of theoretical results.

AASHTO policy uses superelevation to balance the effects of sharper curvature [17]. This balance may be not achieved when emergency steering is needed. This research investigates the radius of curvature of the AVTP with a transient bicycle model duing emergency steering. The result showed that it produces a path radius much sharper than that of the roadway curve when a driver uses an emergency steering maneuver. Particularly, while vehicles are turning from the inside to the outside of the curve, instead, superelevation increases the potential of rollovers. These results suggest support for MacAdam and Bonneson’s conclusion that the vehicle stability on a curve should be analyzed with a complex model [20,21]; once driver behaviors change the forces on vehicles, it may cause rollovers or skids when the lateral acceleration is below the lateral acceleration recommended by the point-mass model.

DIESM could easily lead the driver to make mistakes and cause rollover crashes [18,19]. The present study gave the mathematical relationship between the radius of curvature of the AVTP and the steering wheel instantaneous input and obtained the maximum safe input of the steering wheel against rollovers; the calculation is convenient and the results are accurate. It provides useful insights for transportation engineers and policy makers can identify relationships between emergency steering and truck rollover potential to improve driver training and education programs designed to modify risky driving habits. Furthermore, it provides a theoretical basis for the planning of obstacle avoidance paths of intelligent driving.

Based on the results of this study, it is recommended that more attention should be given to the change of the vehicle travel path when emergency steering, especially for low rollover threshold and high speed vehicles. These findings may provide some references for highway management departments, for example, driving speed of large trucks with low rollover thresholds should be strictly controlled on road sections with frequent crashes, and emergency steering and emergency lane change maneuvers need to be limited for these vehicles.

## 7. Conclusions

The objective of this study was to quantify the truck rollover potential under DIESM. The transient bicycle model was used to develop the mathematical relationship between the radius of curvature of the AVTP and the steering wheel instantaneous input. By the mathematical relationship, the vehicle’s lateral acceleration under DIESM was calculated, and the vehicle’s rollover thresholds were calculated by distinguishing two turning conditions. Finally, the rollover margin was the rollover threshold minus the lateral acceleration under DIESM. Moreover, computer simulation verified the accuracy of theoretical results.

The results showed that (1) the larger the steering wheel instantaneous input, the higher is the potential for rollovers, the higher the driving speed, the higher is the potential for rollovers; (2) there is no direct relationship between the effects of DIESM on truck rollovers and the radius of the roadway curve, but the superelevation significantly affects truck rollovers; (3) the worst turning condition is turning from the inside of the curve to the outside of the curve. These results are satisfactory compared with the previous researches.

This study is needed because DIESM increases the driving risk, especially for trucks. This behavior produces a path radius much sharper than that of the roadway curve, however, it has not attracted enough attention. In theoretical research and practical application, this study developed the mathematical relationship between the radius of curvature of the AVTP and the steering wheel instantaneous input under DIESM. This study quantified the effect of DIESM on the truck rollover potential based on an uncomplicated, easy-to-use and highly accurate model, and obtained the maximum safe instantaneous input of the steering wheel against rollover for trucks on horizontal curves. It is important for drivers to understand the safety implications of the instantaneous emergency steering maneuvers and the extent of the influence of these maneuvers on truck rollover. In general, drivers behave more safely as they are told the risks or consequences of their maneuvers.

However, some aspects were neglected, such as the influence of the deformation of the steering system.

Further research should be directed at collecting information concerning the relative propensity of DIESM to determine if these should be considered in models. Naturalistic driving studies may provide the opportunity to collect these data from equipment installed on vehicles.

## Figures and Tables

**Figure 1 ijerph-19-02025-f001:**
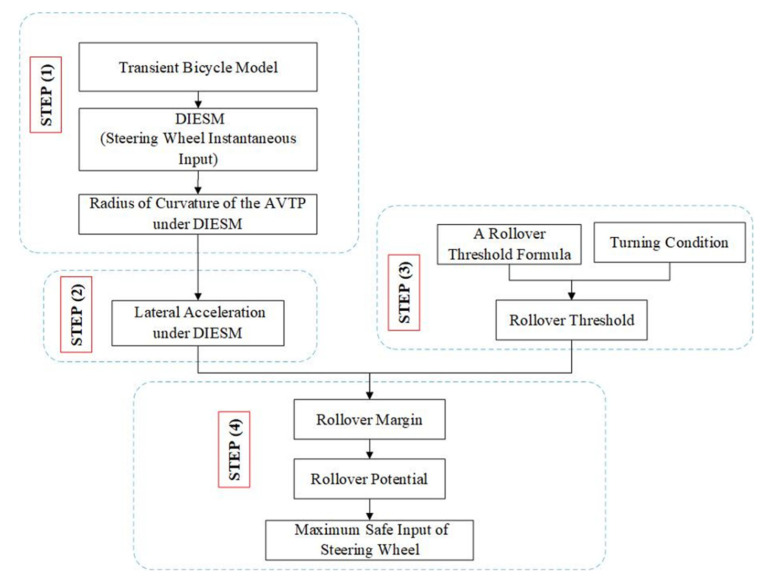
Procedure for evaluating rollover potential.

**Figure 2 ijerph-19-02025-f002:**
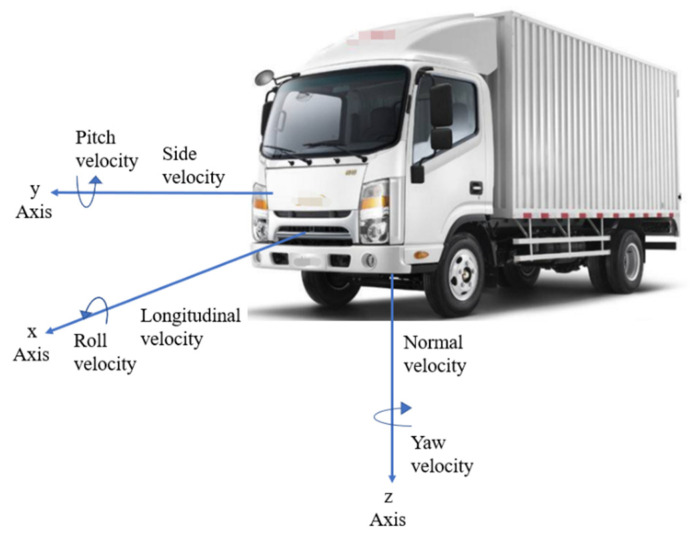
SAE coordinate system.

**Figure 3 ijerph-19-02025-f003:**
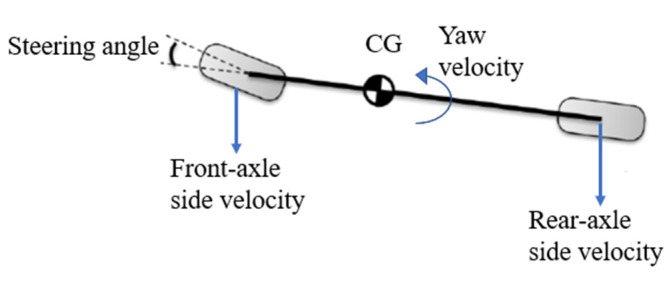
Motion on the xoy plane.

**Figure 4 ijerph-19-02025-f004:**
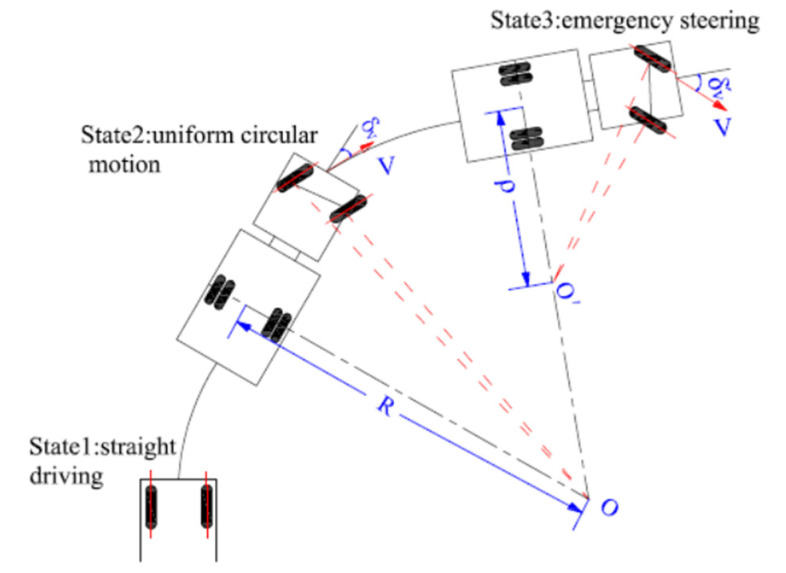
Schematic diagram of the principle of vehicle steering.

**Figure 5 ijerph-19-02025-f005:**
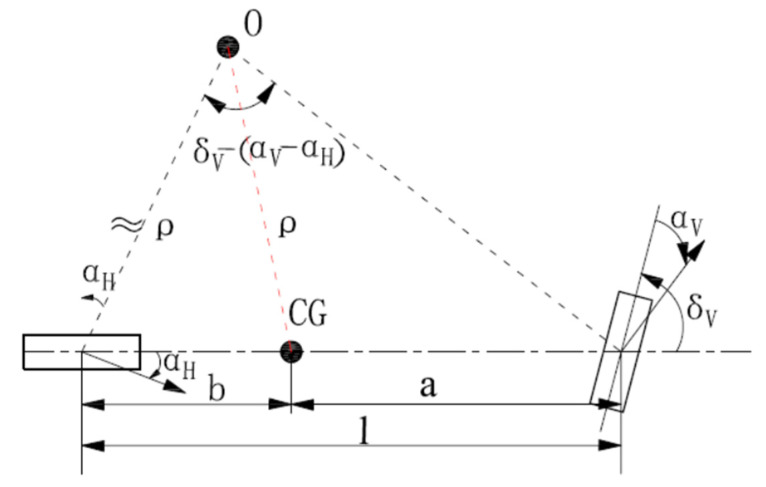
Transient bicycle model at high speeds. Notes: ‘a’ represents the distance from CG to front axle, ‘b’ represents the distance from CG to rear axle.

**Figure 6 ijerph-19-02025-f006:**
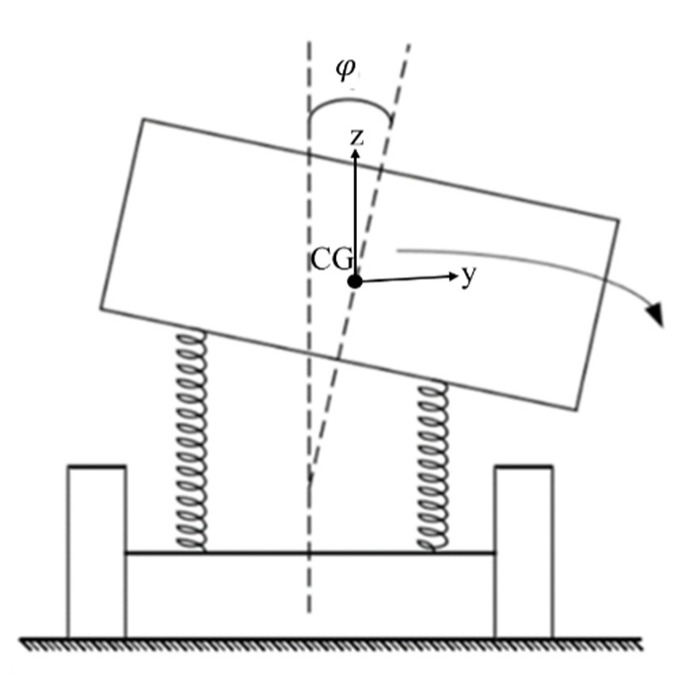
Motion on the yoz plane.

**Figure 7 ijerph-19-02025-f007:**
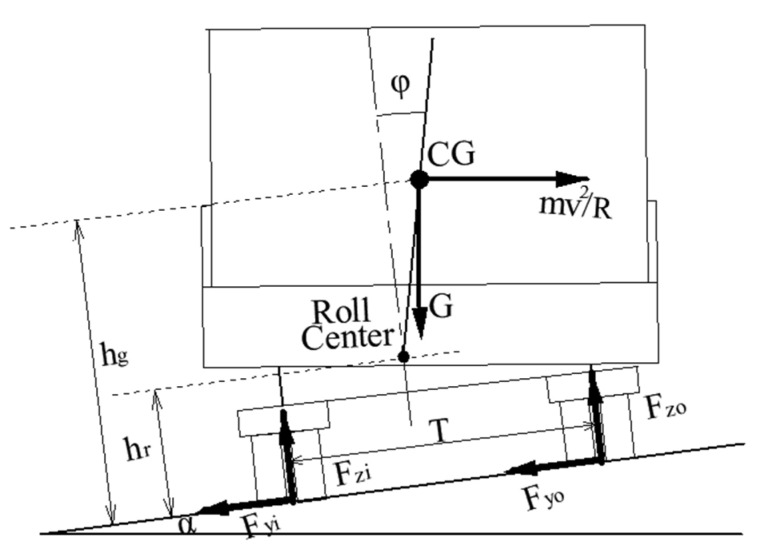
A suspension-vehicle model.

**Figure 8 ijerph-19-02025-f008:**
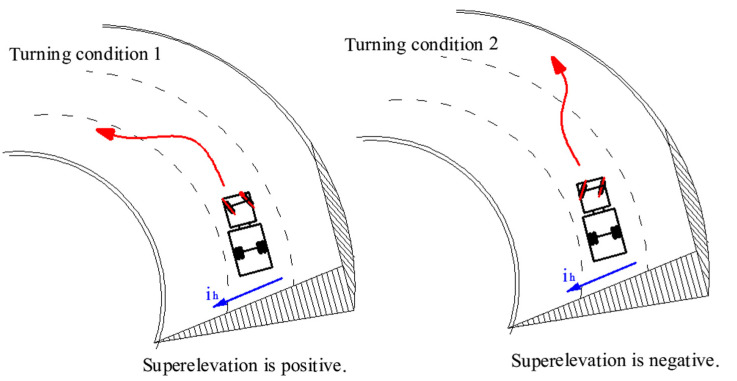
Schematic diagram of two turning conditions.3.3.3. Rollover Margins Based on Lateral Acceleration.

**Figure 9 ijerph-19-02025-f009:**
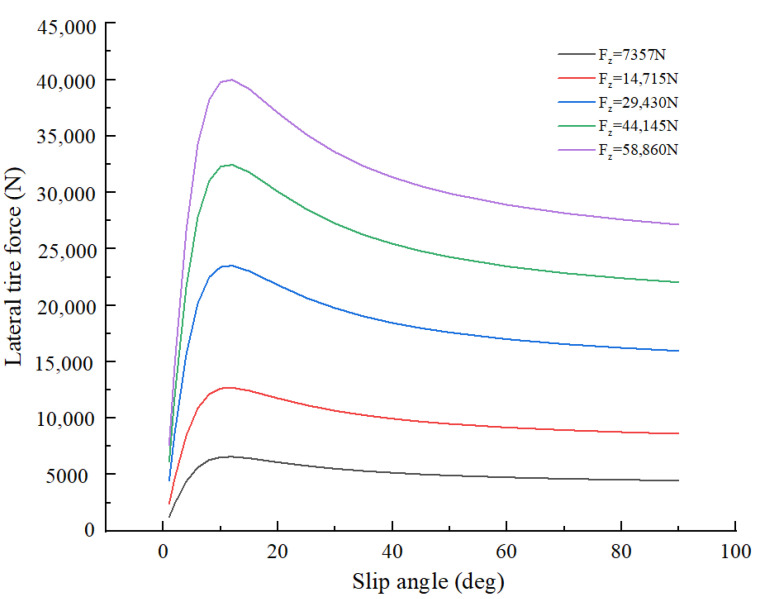
Truck tire curves data sets. Note: ‘F_z_’ represent the vertical loads.

**Figure 10 ijerph-19-02025-f010:**
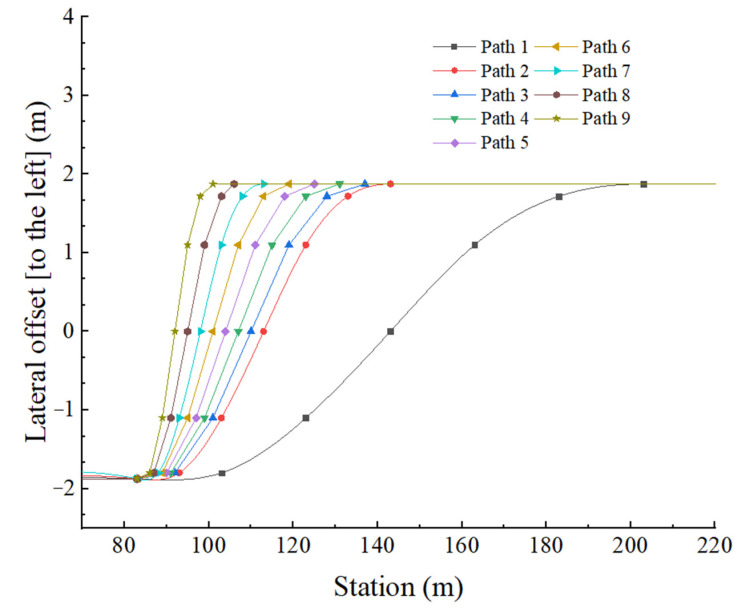
Path coordinate parameters.

**Figure 11 ijerph-19-02025-f011:**
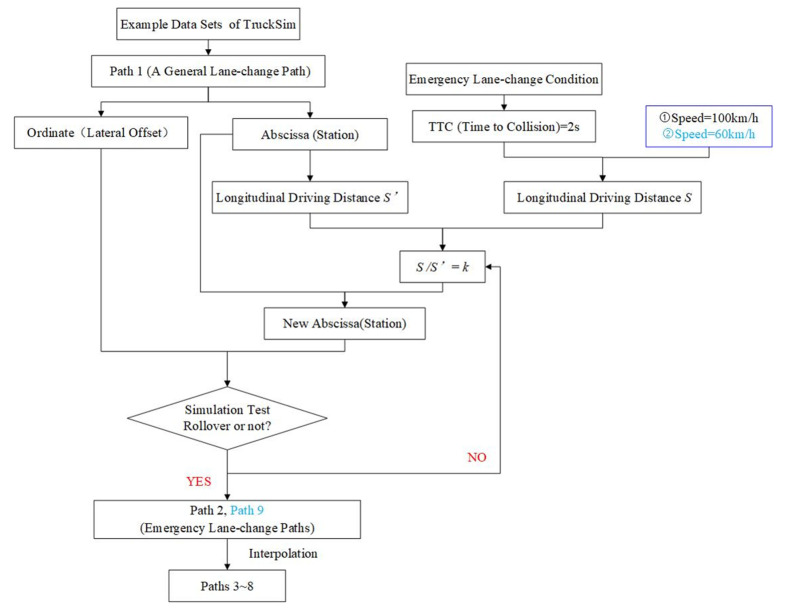
Process of establishing paths 1~9.

**Figure 12 ijerph-19-02025-f012:**
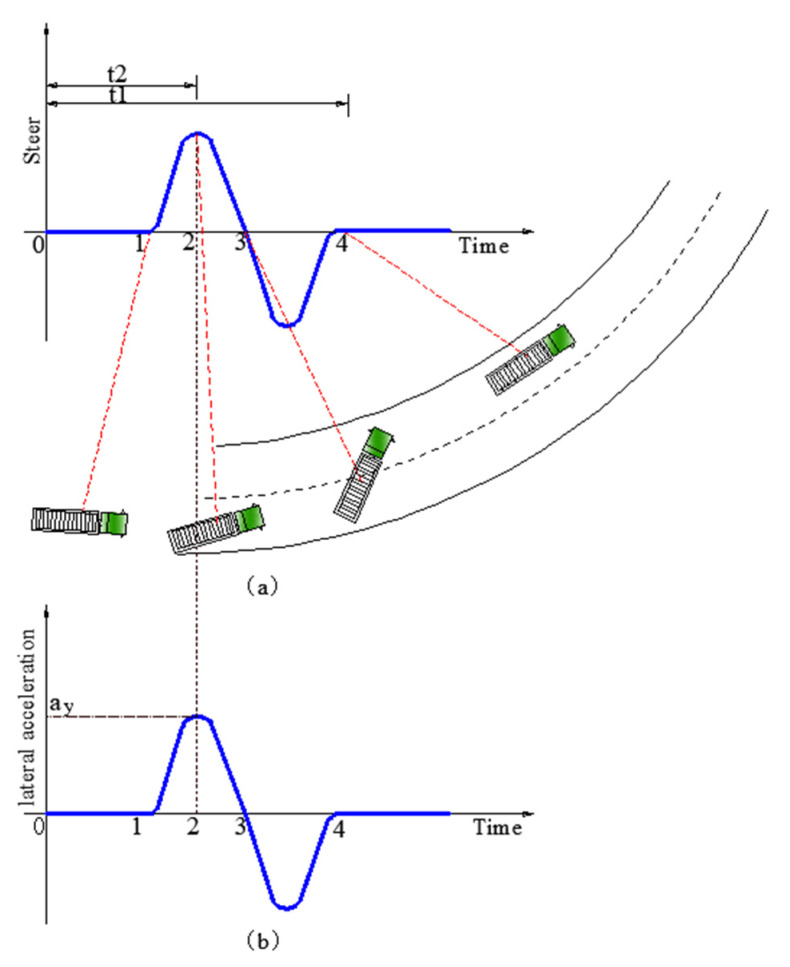
Lane-change simulation. (**a**) Steering inputs and vehicle location on the roadway VS time; (**b**) Lateral acceleration VS time.

**Figure 13 ijerph-19-02025-f013:**
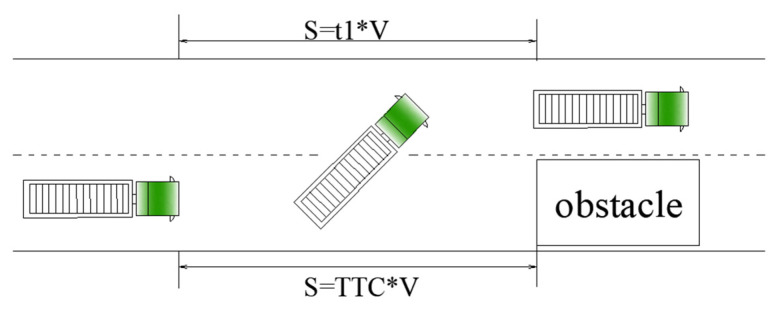
Diagram of obstacle avoidance at high speeds in an emergency.

**Figure 14 ijerph-19-02025-f014:**
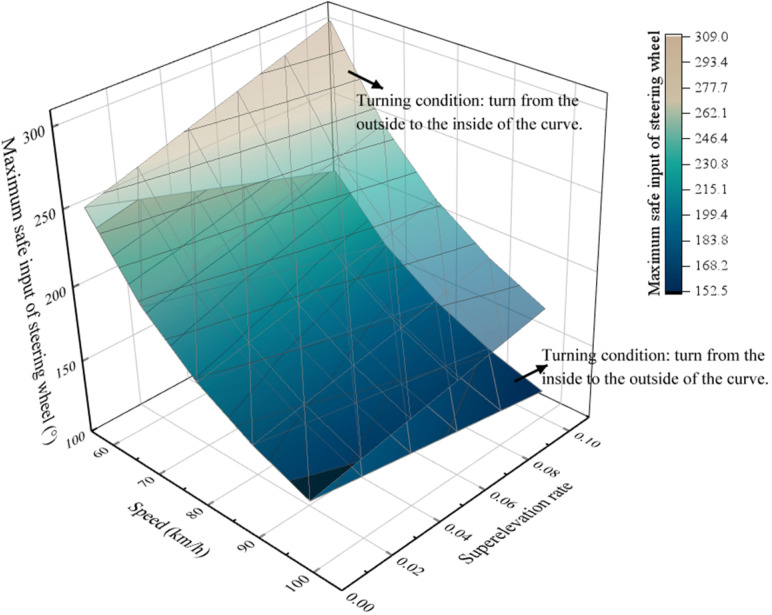
Relationship between speed, superelevation rate and maximum safe input of the steering wheel.

**Figure 15 ijerph-19-02025-f015:**
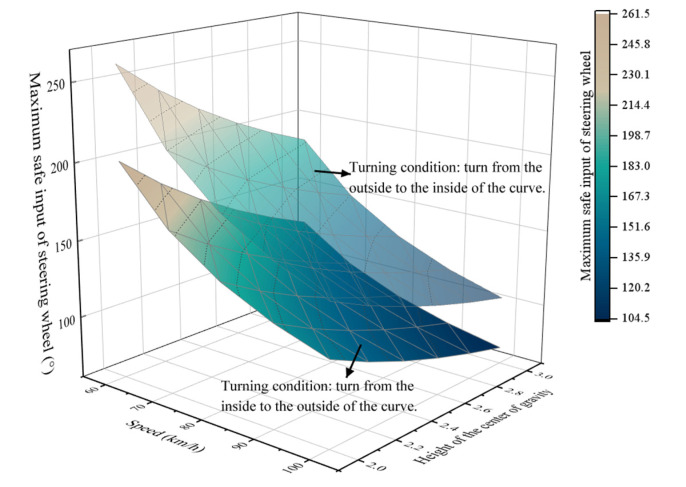
Relationship between speed, CG height and maximum safe input of the steering wheel.

**Figure 16 ijerph-19-02025-f016:**
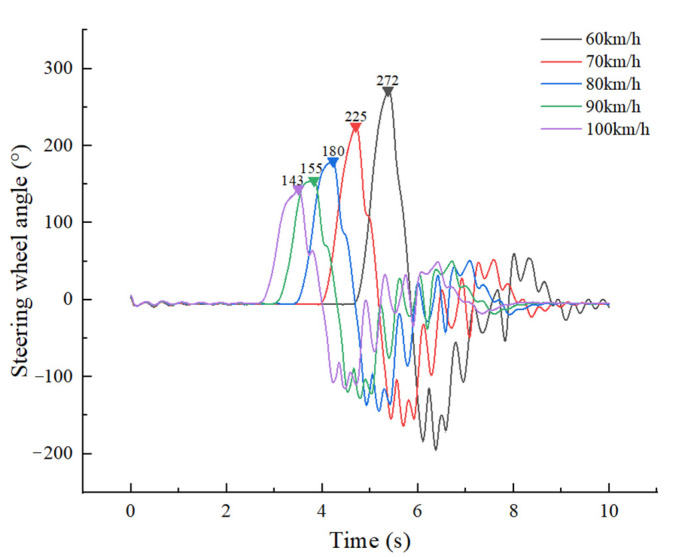
Steering wheel angle input vs. time.

**Figure 17 ijerph-19-02025-f017:**
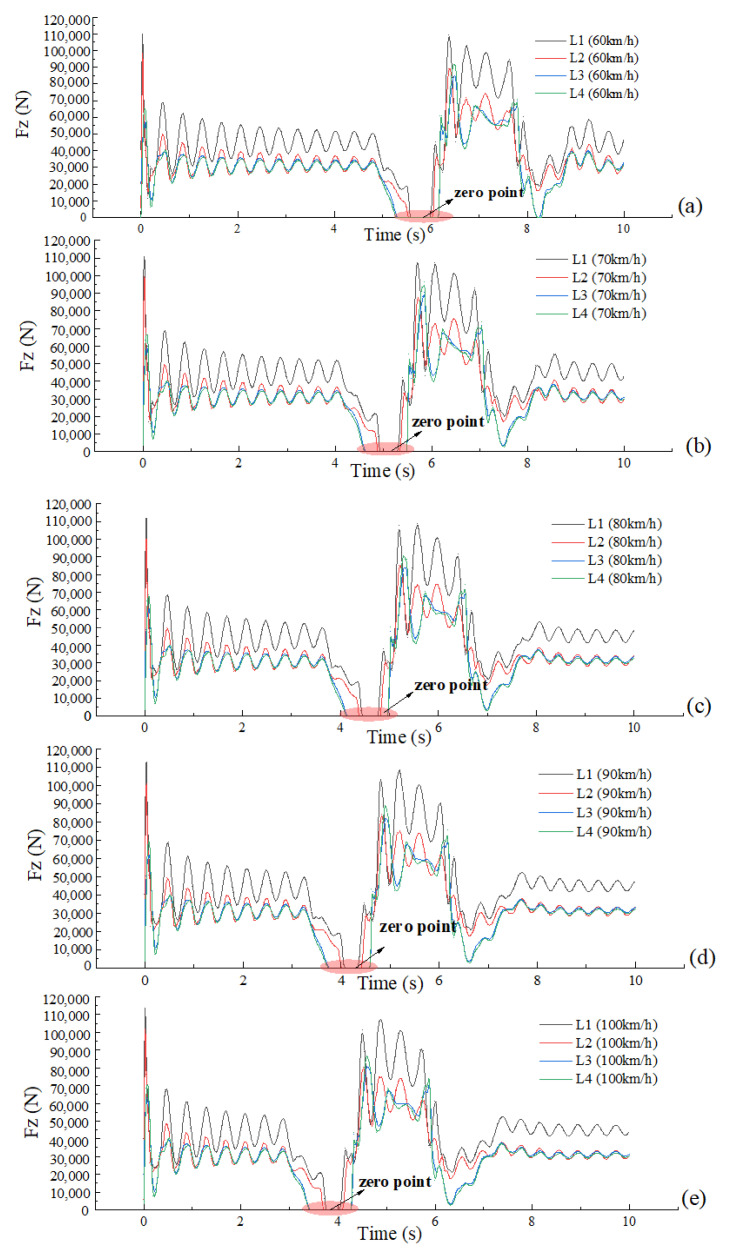
Vertical force of left tire VS time. (**a**) V = 60 km/h, (**b**) V = 70 km/h, (**c**) V = 80 km/h, (**d**) V = 90 km/h, (**e**) V = 100 km/h.

**Table 1 ijerph-19-02025-t001:** Driver behavior elements of scenes.

Scene ID	Driver Behavior	
	Speed	Lane-Change Path
	(km/h)	Number
1	100	2
2	100	3
3	100	4
4	100	5
5	100	6
6	100	7
7	100	8
8	100	9
9	90	2
10	90	3
11	90	4
12	90	5
13	90	6
14	90	7
15	90	8
16	90	9
17	80	2
18	80	3
19	80	4
20	80	5
21	80	6
22	80	7
23	80	8
24	80	9
25	70	2
26	70	3
27	70	4
28	70	5
29	70	6
30	70	7
31	70	8
32	70	9
33	60	2
34	60	3
35	60	4
36	60	5
37	60	6
38	60	7
39	60	8
40	60	9

**Table 2 ijerph-19-02025-t002:** Comparison of theoretical and simulation results.

Driver Behavior		Simulation Results	Theoretical Calculation Results	Difference
Speed	Lane-Change Path			
(km/h)	Number	(°)	(°)	(%)
100	4	143	147	2.7
90	5	155	162	4.3
80	6	180	182	1.1
70	7	225	212	6.1
60	8	272	259	5.0

*Notes:*colum11=|column9−column10|/column10.

## Data Availability

The data used to support the findings of this study are included within the article.

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
