# Peer review of "Study on the Maximum Safe Instantaneous Input of the Steering Wheel against Rollover for Trucks on Horizontal Curves"

_ijerph, 2022, doi:10.3390/ijerph19042025_

Round 1

Reviewer 1 Report

In the literature review, refer to research in this area, presented in the International Journal of Environmental Research and Public Health.

In order to improve the quality of the article, I send additional comments to the reviewed article: The authors of the study presented an interesting approach to the analysis of vehicle stability on horizontal curves. The problem is current and requires research in this area. The work presents the introduction, context and methodology of the research well. The results are presented clearly although they require additional comment. Below are some remarks to the presented content of the work:

  1. Extend the literature review, in particular refer to other publications in this area presented in the International Journal of Environmental Research and Public Health.
  2. To emphasize the approach of the authors of the work to those already existing in the literature and to show the differences.
  3. In the analyzes, the authors showed that there are differences in the results of theoretical and simulation studies (Table 2.). The difference of the maximum safe inputs of steering wheel based on theory and simulation was referred to for a speed of 70 km/h which is 6.1%. There are no analyzes for the remaining driving speeds, moreover, the maximum differences are different for different speeds - is there a relationship between these differences in the function of vehicle speed - please comment on this.
  4. Figure 17 d: incorrect description - the simulation applies to a speed of 90 km / h on the L3 graph (60 km / h).
  5. In the discussion of the research, the authors presented general conclusions without referring to the quantitative data obtained in the research.

The proofreading of the thesis will increase its substantive value and the quality of the publication.

Reviewer 2 Report

The topic analyzed in the manuscript is interesting and relevant; however, some important notes are addressed to authors.

The main purpose of the research is not highlighted enough. What is the theoretical or practical application of the research results?

The achieved results of speed and super-elevation influence on  cornering stability (steering input) is too general.

Line 46: what kind of “design speed” authors mean – vehicle design speed or road design speed? (the same at line 100).

It is about 0.35g at lines 114-115?

The thought “…all vehicles have suspension systems” (line 120) is too general. It is recommended better to pay attention for different kind of suspensions or its specific for trucks.

The sentence “Some studies showed…” (lines 130-135) is to long and complicated to read and understand.

Only one word “given shown” should be in line 160.

Line 189-197 described vehicle steering states are very general, usually described in all handbooks of vehicle dynamics. It is recommended to decline this part or modify. Moreover, a dot in line 193 is not necessary.

There are some text corrections needed after equations; e.g.: “° ; m;” (line 210), “axle, m;?-distance” (line 225), “input, °??- steering” (line 231). Fing marked in attached pdf.

No space left after Fig. 2, 3, 4 (above), 5, 6, 7, 8 (above as well), 10, 11, 12, 13 (above), 14, 15 (above).

Lateral acceleration eq. (9) according to its physical meaning is better to name as a centrifugal acceleration.

 It is usual to use 3-DOF or 2-DOF (line 260, 260).

Coordinate axis is necessary in Fig. 6.

Please explain why no axle roll is added (Fig. 6)? Trucks usually have large tire deformation during sharp cornering, which contributes to total roll angle of the vehicle.

Why speed v is switched to capital from eq. (18).

What are the vertical loads for tire characteristic (Fig. 9)? It is better to draw a graph instead of making a printscreen from the software.

Why Magic Formula is used to get stiffness from its characteristic? It is only a gradient (slope) at initial angles.

If a complete tire characteristic is known from TruckSim softweare, there is no need to use a Magic Formula.

“As the path is given, steering information is given.” – is steering given or calculated? Please explain simulation conditions more detail.

There is no need to put the data in Table 1, which is constant per all ID’s, as: curve radius, i_h, I, miu, total mass, CG height. The table now looks too crowded. This information should be better mentioned in text. The same for Table 2.

 It is difficult to read the axis in horizontal plane of Fig. 14.

 “if the truck turns from the inside to the outside of the curve at a constant speed, the maximum safe input of steering wheel decreases significantly with the increase of super-elevation rate” – it is well known that road super-elevation increases vehicle cornering stability and allows to keep cornering with higher speed; could you explain your sentence at this context (line 500-502).

What about the rate of steering wheel angle (Fig. 16)? Is it the same for all cases?

The size of Fig. 17 is too small and difficult to read.

Why only tire stiffness was used for theoretical calculations instead of full tire characteristics available from TruckSim? Maybe this can be one of the cause for obtained differences between theoretical and simulation results?

Round 2

Reviewer 2 Report

Thank you for your answers and manuscript corrections.